# Nonlinear and Multidelayed Effects of Meteorological Drivers on Human Respiratory Syncytial Virus Infection in Japan

**DOI:** 10.3390/v15091914

**Published:** 2023-09-12

**Authors:** Keita Wagatsuma, Iain S. Koolhof, Reiko Saito

**Affiliations:** 1Division of International Health (Public Health), Graduate School of Medical and Dental Sciences, Niigata University, Niigata 951-8510, Japan; jasmine@med.niigata-u.ac.jp; 2Japan Society for the Promotion of Science, Tokyo 102-0083, Japan; 3College of Health and Medicine, School of Medicine, University of Tasmania, Hobart 7000, Australia; iain.koolhof@health.tas.gov.au

**Keywords:** human respiratory syncytial virus, meteorological drivers, transmission dynamics, epidemics, Japan

## Abstract

In this study, we aimed to characterize the nonlinear and multidelayed effects of multiple meteorological drivers on human respiratory syncytial virus (HRSV) infection epidemics in Japan. The prefecture-specific weekly time-series of the number of newly confirmed HRSV infection cases and multiple meteorological variables were collected for 47 Japanese prefectures from 1 January 2014 to 31 December 2019. We combined standard time-series generalized linear models with distributed lag nonlinear models to determine the exposure–lag–response association between the incidence relative risks (IRRs) of HRSV infection and its meteorological drivers. Pooling the 2-week cumulative estimates showed that overall high ambient temperatures (22.7 °C at the 75th percentile compared to 16.3 °C) and high relative humidity (76.4% at the 75th percentile compared to 70.4%) were associated with higher HRSV infection incidence (IRR for ambient temperature 1.068, 95% confidence interval [CI], 1.056–1.079; IRR for relative humidity 1.045, 95% CI, 1.032–1.059). Precipitation revealed a positive association trend, and for wind speed, clear evidence of a negative association was found. Our findings provide a basic picture of the seasonality of HRSV transmission and its nonlinear association with multiple meteorological drivers in the pre-HRSV-vaccination and pre-coronavirus disease 2019 (COVID-19) era in Japan.

## 1. Introduction

The human respiratory syncytial virus (HRSV) infects the human respiratory tract, causing clinically severe pneumonia in young children and bronchitis in infants [1]. Globally, HRSV-induced acute lower respiratory tract infections (ALRIs) are responsible for approximately 70,000 deaths in children under the age of 5 years and hospitalization of approximately 3.4 million people [2,3]. The magnitude of the global disease burden caused by HRSV has recently been recognized to affect not only infants and young children but also the elderly (≥65 years), and it remains a significant public health concern [4,5]. The transmission dynamics of HRSV infection epidemics are seasonally driven in temperate regions, with most infections occurring during annual autumn–winter (September–February) seasons; however, a seasonal shift in the HRSV infection epidemics to the summer–autumn (June–November) seasons between 2017 and 2019 has been reported in Japan [6,7,8,9]. For instance, epidemics of HRSV infection in the United States and 15 European countries, where the climatic environment is like that of Japan, are also concentrated in autumn–winter seasons [6,10]. It should be noted that off-season HRSV infection epidemics were observed during the coronavirus disease 2019 (COVID-19) pandemic in Japan (i.e., no major epidemic in 2020, and an unusually high number of cases reported in 2021) [11,12].

The transmission dynamics of HRSV are shaped by a web of biological and nonbiological factors, and it has been suggested that in addition to intrinsic transmission factors such as innate immunity, viral survival, and antigenic variation, extrinsic seasonal climate-driven factors influence the transmission dynamics [13,14]. Although previous studies have reported partially conflicting results, several phenomenological trends in the relative role of multiple meteorological conditions (e.g., mean ambient temperature, relative humidity, precipitation, and wind speed) in HRSV transmission have emerged. For instance, two systematic reviews concluded that high ambient temperature and relative humidity were associated with HRSV transmission in tropical and subtropical regions, while in temperate regions, low ambient temperature and high relative humidity were associated with the occurrence of several HRSV-caused epidemics [15,16]. In other epidemiological studies, ambient temperature has been found to have a negative association with HRSV infection incidence in European countries [17,18], the United States [19,20], and Mainland China [21,22,23,24]. Meanwhile, relative humidity has been reported to have a positive association with HRSV infection incidence in European countries and Hong Kong [17,25,26]. Overall, few epidemiological studies have described the relationship between precipitation and wind speed and HRSV infection incidence; however, an association between peak precipitation and HRSV infection incidence has been observed in several tropical regions [27,28,29,30,31], and a moderate negative correlation between wind speed and its incidence has been observed in Kenya [32]. Indeed, the differences in these reports could partially be explained by the variation in the spatial scales of analysis, application of different statistical methods, and consideration of various potential confounders. Epidemiologic studies assessing the relationship between climate variability and transmission could have methodologic limitations that may introduce bias and limit causal inferences [33,34,35].

Although previous studies have gradually revealed the associations between the climate variability and the transmission dynamics of HRSV, the effect of meteorological drivers on the timing and intensity of its epidemics is poorly understood. Generally, the effects of climate variability and disease transmission are complex, nonlinear, and often shape multiple, delayed events, limiting statistical inferences based on traditional deterministic linear modelling and single-delayed effects frameworks [36]. While these traditional methods aid in the forecasting of HRSV infection epidemics, current modelling frameworks may oversimplify the nature of relationships between meteorological drivers and disease dynamics. The time-varying distributed lag nonlinear models (DLNMs) developed in the last decade have been relatively underused, despite being well suited for time-series statistical modelling of infectious disease dynamics [37,38]. In recent years, these models have begun to be adapted to climate-sensitive vector-borne diseases (e.g., dengue fever, malaria, and West Nile virus) [39,40,41,42,43,44], but their application for respiratory virus infections, including HRSV infections, has been quite limited. In the present study, we aimed to explore the nonlinear and multidelayed effects of multiple meteorological drivers (i.e., mean ambient temperature, relative humidity, precipitation, and wind speed) on the seasonality of HRSV infections using weekly surveillance data across 47 prefectures in Japan over the 2014–2019 period by designing a cutting-edge time-series statistical model. These exposure–lag–response associations can reveal how meteorological drivers affect HRSV infection risk in the weeks leading up to an outbreak. Furthermore, if the explanatory powers of these meteorological drivers can be determined, the seasonality and transmission of these viruses in different environments can be better understood, which would contribute to the development of early warning systems to respond in a timely manner to annual HRSV epidemic surges and public health strategies and interventions. By characterizing the relative role of these meteorological drivers on HRSV infection dynamics, we attempted to gain an insight into the seasonal mechanisms of the disease, thereby rendering basic information to disentangle its extrinsic environmental drivers.

## 2. Materials and Methods

### 2.1. Design Setting

This time-series statistical modeling analysis investigated the potential nonlinear and multilagged relationships of HRSV transmission with multiple meteorological drivers (i.e., mean ambient temperature, relative humidity, rainfall, and wind speed) in Japan, using an ecological study design. In this study, we systematically collected and analyzed epidemiological data for the number of weekly newly confirmed HRSV infection cases and meteorological drivers (i.e., mean ambient temperature, relative humidity, rainfall, and wind speed) from 1 January 2014 to 31 December 2019 (between the 1st week of 2014 and 52nd week of 2019) across 47 prefectures in Japan, covering the entire country. We selected these locations based on the geographical diversity and availability of data across the study period. It should be noted that the transmission dynamics of HRSV have drastically changed since 2020 owing to the COVID-19 pandemic; hence, this period was not included in the analysis [11,12]. The post-COVID-19 pandemic period will be evaluated in a future project. Japan is located at latitudes of approximately 26–43° N and longitudes of approximately 127–141° E in the Western Pacific Region and comprises a total of 47 Japanese prefectures (Appendix A). Given their geoclimatic characteristics, 46 prefectures, excluding Okinawa, are classified as temperate regions, and Okinawa as a subtropical region with four distinct seasons in Japan; summers are hot and humid (June–August), while winters are cold and dry (December–February).

### 2.2. Empirical Datasets

#### 2.2.1. National HRSV Surveillance Data

The HRSV infection epidemiological data used in this study were obtained from the Infectious Disease Weekly Report (IDWR), sourced from the National Epidemiological Surveillance of Infectious Diseases (NESID) data published by the National Institute of Infectious Diseases, Japan (NIID), under the authority of the Ministry of Health, Labor and Welfare, Japan (MHLW) [45]. The MHLW manages approximately 3000 pediatric sentinel sites (i.e., hospitals and clinics) in Japan and reports the number of patients diagnosed with HRSV infection on a weekly basis to the prefecture or municipal public health sectors in Japan [46,47]. A confirmed case of HRSV infection is defined as a positive result in a rapid diagnostic test (RDT) using a test kit licensed in Japan or a laboratory confirmation such as virus isolation or antibody titer increase in paired sera according to the MHLW guidelines [48]. The number of sentinels assigned to each public health service area is based on population size: a public health center with <30,000 individuals is assigned one sentinel, a center with 30,000–75,000 individuals, two sentinels, and that with >75,000 individuals, three or more sentinels, as determined by the following formula: 3  +  (population − 75,000)/50,000 [9]. These sentinel sites forward clinical data to approximately 60 prefectural or municipal public health sectors, and the data are electronically reported to the NIID; the number of HRSV infection cases is released weekly on the NIID website. In the present study, we extracted the number of weekly newly confirmed HRSV infection cases across all 47 prefectures in Japan reported in weeks 1–52 from 2014 to 2019, from the NESID database.

#### 2.2.2. Meteorological Data

The Automated Meteorological Data Acquisition System (AMeDAS) developed by the Japan Meteorological Agency is a high-resolution surface observation network for investigating meteorological conditions in Japan. In the present study, we retrieved prefecture-specific daily time-series of meteorological data across 47 prefectures in Japan collected at the AMeDAS over the study period [49]. The weekly meteorological data, including mean ambient temperature in degrees Celsius (°C), relative humidity in percent (%), precipitation in millimeters (mm), and wind speed in meters per second (m/s), published by the website were calculated from the daily records and utilized as independent variables in the time-series statistical models presented here. These meteorological variables were chosen according to their availability and use in previous HRSV modelling studies [8,11,17,19,50,51,52]. Meteorological data collected from meteorological observatories (meteorological stations) situated in the prefectural capitals were utilized for each prefecture.

### 2.3. Statistical Analysis

#### 2.3.1. Descriptive Statistics

To determine the key characteristics of the multitudes of time-series datasets used in this study, we described the probability distributions of the number of weekly newly confirmed HRSV infection cases and meteorological drivers (i.e., mean ambient temperature, relative humidity, precipitation, and wind speed) across all 47 prefectures in Japan during the study period by utilizing the following descriptive statistics: mean, standard deviation (SD), minimum (Min), 25th percentile (P_25_), 50th percentile (P_50_), 75th percentile (P_75_), and maximum (Max).

#### 2.3.2. Construction of the Time-Series Statistical Model

To establish a robust and reliable time-series statistical model, multiple stages were incorporated into it [11,53,54]. Prior to constructing the model, we checked the probability distribution of the dependent variables and number of weekly newly confirmed HRSV infection cases (the normality of probability distribution was assessed by the Shapiro–Wilk test) (Appendix A), followed by an assessment of the relationships (e.g., linearity) between number of weekly newly confirmed HRSV infection cases and each independent variable. Generally, by assessing the transformation of dependent variables in time-series-driven systems, these variables can be used as stationary effects, often improving forecasting accuracy. All dependent variables and independent variables included in the statistical models were assessed for multicollinearity using pairwise Spearman’s rank-order cross-correlation coefficient (*ρ*). If the variables were found to be strongly linearly correlated (cut-off of |*ρ*| > 0.8), the variable with the largest mean absolute statistical correlation with the other independent variables was eliminated [55]. In the preliminary analysis, no independent variables showing strong statistical linear correlations were observed (Appendix A).

In the present study, we fitted standard time-series multivariate generalized linear models (GLMs) with a quasi-Poisson distribution family and logarithmic-link function, allowing for overdispersion in the observational data, by adding the time-varying DLNMs to simultaneously describe nonlinear and multidelayed dependencies between the incidence relative risks (IRRs) of HRSV infection and meteorological drivers. Briefly, a DLNM was designed for each targeted meteorological driver and added to the regression equation of GLM. This method was used as a primary model to disentangle the underlying complex association between the number of weekly newly confirmed HRSV infection cases and different multiple meteorological drivers (i.e., mean ambient temperature, relative humidity, precipitation, and wind speed) as main exposures. In a DLNM, a cross-basis function that combines two tensor product of basis-functions is introduced: one represents the dependent variable’s probability distribution in the independent variable and lagging dimensions to simultaneously assess the lag effect and the other represents the nonlinear effects of the exposure driver [37,38,56].

The cross-basis term for time-varying DLNMs acts as basis function predictor in two dimensions: exposure and lag spaces. Specifically, we modeled exposure–response associations using a natural cubic spline with three degrees of freedom (df) and modeled the lag–response association using a natural cubic spline with three df. To adjust for the possible harvesting and misalignment from the delayed heterogeneity of meteorological drivers of HRSV infection dynamics, based on multiple accumulated studies investigating the incubation period (i.e., approximately 2–8 days) and reporting delays of HRSV [8,17,19,50,51,52,57,58], we considered temporal lags (i.e., delays in potential effect) of up to 2 weeks as the default lag structure for the cross-basis function of each independent variable related to the dependent variable. These temporal time lags are useful for describing the biologically and physically plausible time lags in the population dynamics, natural history of the HRSV host reservoir, and subsequent incubation period before disease notification.

The general algebraic definition of the time-series statistical models is formulated as follows:$$\begin{array}{ll} \log\left[\left(y_{\left[i,t\right]}|r_{\left[i,t\right]}\right)\right] & :=\alpha_{\left[0\right]}+\sum_{s}\beta_{\left[s\right]}f_{g}\left(x_{\left[i,t,s\right]};df\right)+\sum_{u}h_{g}\left(z_{\left[i,t,u\right]};\theta \right)+e_{\left[t\right]}+p_{\left[t\right]}\\ & +s_{\left[t\right]}+\log\left[r_{\left[i,t-1\right]}\right]+\log\left[O_{\left[t\right]}\right]+\varepsilon_{\left[i,t\right]} \end{array}$$
where $y_{\left[i,t\right]}$ is the outcome time-series; $r_{\left[i,t\right]}$ is the expected time-series of the number of weekly newly confirmed HRSV infection cases in prefecture *i* on week *t*; the term $\alpha_{\left[0\right]}$ corresponds to the overall intercept; $\sum_{s}\beta_{\left[s\right]}f_{g}\left(x_{\left[i,t,s\right]};df\right)$ denotes the cross-basis function of DLNMs with exposure and multilagged effects modelled by a natural cubic spline function and a linear function of multiple meteorological drivers (i.e., mean ambient temperature, relative humidity, precipitation, and wind speed) in prefecture *i* in week *t*, respectively. We also modeled baseline risk along with shared long-term seasonal variations and cycles and short-term trends by incorporating natural cubic splines of time (7 degrees of freedom (df) per year) as term $\sum_{u}h_{g}\left(z_{\left[i,t,u\right]};\theta \right)$, year as term $e_{\left[t\right]}$, and number of public holidays per week as term $p_{\left[t\right]}$, with the fixed effects variables as possible confounders [11,53,54,59,60,61]. $s_{\left[t\right]}$ denotes prefectural characteristics or regional variable indicators in prefecture *i*. Furthermore, the autocorrelation of residuals in the case of infectious disease was pathogen-specific and needed to be accounted for; therefore, autoregressive terms at order one (as term $r_{\left[i,t-1\right]}$) was incorporated into the statistical models, accounting for potential serial correlation [53,54,59,60,61]. The teem $\log\left[O_{\left[t\right]}\right]$ denotes the logarithm of the yearly population (per 100,000) by prefecture as the offset term [62]. The use of these population estimates for our denominator in the models allowed for the exponentiation of each coefficient to be expressed as an IRR, which is an intuitive representation of the association of variables with the increased risk of HRSV infection incidence. The term $\varepsilon_{\left[i,t\right]}$ indicate errors. To quantify the total contribution, independent effects, and relative importance of multiple meteorological drivers, we included all variables in the same model [53,63]. By including all variables of interest in the same regression equation, we were able to strengthen the interpretation of the effects as independent or additive, based on accumulated empirical knowledge.

To test the sensitivity of the results to the modeling choices described above, we repeated the analysis by varying the df of the natural cubic spline of time from 7 df per year to 3 df or 11 df per year. We also performed a sensitivity analysis of the observed effect on the weeks of lags accounted for in the statistical model by modifying the length of the lag period from 2–3 and 4 weeks. We quantified the estimates as cumulative-week (i.e., 0–2 weeks) and single-week (i.e., 0, 1, and 2 weeks) IRRs together with the 95% confidence intervals (CIs) at the 25th and 75th percentiles of all meteorological drivers to determine the strengths of the associations. The reference value for ambient temperature and relative humidity were set as the median (50th percentile), while the precipitation and wind speed were set as 0.0 mm and 0.0 m/s, respectively. Additionally, we plotted the exposure–response curves to describe the overall cumulative associations. Statistical significance was considered at a *p*-value of <0.05 (i.e., type I error), on a two-tailed test. All analyses were performed using STATA version 15.1 statistical software (Stata Corp, College Station, TX, USA) and R statistical programming software version 4.1.0 (R Foundation for Statistical Computing, Vienna, Austria) using the “dlnm” [64].

### 2.4. Ethical Considerations

The present ecological modeling study analyzed publicly available data in Japan. As such, the epidemiological datasets utilized in this study were de-identified and fully anonymized in advance, and the analysis of publicly available data with no identifying information did not require ethical approval. The present study was conducted in accordance with the Declaration of Helsinki (as revised in 2013).

## 3. Results

### 3.1. Descriptive Analysis

Overall, 721,709 newly confirmed HRSV infection cases were observed during the 313-week study period across all 47 prefectures in Japan (Appendix A). The mean number of the weekly newly confirmed cases, i.e., the average for all prefectures and weeks during the study period, was 49 (range, 0–892) (Table 1). The weekly mean ambient temperature, relative humidity, precipitation, and wind speed also had wide ranges (−6.7–32.2 °C, 30.8–97.2%, 0.0–94.1 mm, and 0.9–11.3 m/s, respectively).

### 3.2. Assessing the Nonlinear Effects of Meteorological Drivers on HRSV Infection Incidence

The pooled overall cumulative relationships between multiple meteorological drivers (i.e., mean ambient temperature, relative humidity, precipitation, and wind speed) and HRSV infection incidence are illustrated in Figure 1. Notably, we observed a nonlinear relationship between mean ambient temperature and HRSV infection incidence (Figure 1A). In particular, for the range of approximately 10.0–15.0 °C, we found a linear inverse mean ambient temperature–incidence relationship, with low mean ambient temperatures associated with increased HSRV infection incidence. In contrast, the cumulative IRR increased when the mean ambient temperature was dramatically higher than approximately 20.0 °C. Specifically, the corresponding cumulative IRRs were 1.023 (95% CI, 1.008–1.039) at 8.1 °C (25th percentile), and 1.068 (95% CI, 1.056–1.079) at 22.7 °C (75th percentile), respectively, with reference to the IRR at 16.3 °C (Table 1). A J-shaped association was observed between relative humidity and HRSV infection incidence (Figure 1B). Relative humidity exhibited a positive and almost linear relationship with HRSV infection incidence above approximately 70.0%. The 2-week cumulative IRRs were 0.997 (95% CI, 0.971–1.003) at a relative humidity of 63.1% (25th percentile) and 1.045 (95% CI, 1.032–1.059) at a relative humidity of 76.4% (75th percentile) with reference to the IRR at 70.4% (Table 1). The overall relationship between precipitation and HRSV infection incidence was weak (Figure 1C). An almost nonlinear positive association trend with incidence was observed across the entire precipitation range. For instance, the 2-week cumulative IRRs were 1.027 (95% CI, 1.015–1.038) at a wind speed of 0.7 mm (25th percentile) and 1.064 (95% CI, 1.032–1.097) at 6.2 mm (75th percentile) with reference to that at 0.0 mm (Table 1). A clear inverse association was observed between wind speed and HRSV infection incidence (Figure 1D). Specifically, the cumulative IRR decreased rapidly with increasing wind speed across the entire range. The 2-week cumulative IRRs were 0.793 (95% CI, 0.697–0.903) at a wind speed of 2.2 m/s (25th percentile) and 0.775 (95% CI, 0.678–0.885) at 3.4 m/s (75th percentile), with reference to that at 0.0 m/s (Table 1).

### 3.3. Sensitivity Analysis

The main findings described above were confirmed by repeating the series of sensitivity analyses utilizing alternative DLNM specifications. For all meteorological drivers (i.e., mean ambient temperature, relative humidity, precipitation, and wind speed), changing the incorporation of the df of the natural cubic spline of time from 7 df per year to 3 df (Appendix A) or 11 df per year (Appendix A) revealed that the observed risk effect shape was substantially robust over the different parameterizations. Generally, although we observed wider 95% CIs after slightly increasing the df for each of the meteorological drivers, the estimated shapes of the exposure-response functions remained largely similar. Further, for all meteorological drivers, on changing the incorporation from 0–2 lag weeks to 0–3 lag weeks (Appendix A) and 0–4 lag weeks (Appendix A), the observed shapes of the incidence curves were found to be consistent. Generally, model uncertainties tend to increase when more weeks are included. Taken together, the series of sensitivity analyses confirmed the robustness of the findings of the main analysis.

## 4. Discussion

In the present study, we retrospectively modelled the nonlinear and multidelayed temporal effects of multiple meteorological drivers (i.e., mean ambient temperature, relative humidity, precipitation, and wind speed) on HRSV seasonality using multiseason data across all 47 prefectures in Japan. To our knowledge, this study presents one of the most comprehensive assessments of the effects of meteorological drivers on HRSV infection seasonality across a large gradient of climate conditions related to latitudes and longitudes. Overall, we found evidence of a modest nonmonotonic association between these meteorological drivers and HRSV infection incidence over lags of 0–2 weeks, contributing to approximately 14.8% of the variation in IRR over the study period. More specifically, an increase in mean ambient temperature from 8.1 °C (25th percentile) to 22.7 °C (75th percentile) was associated with an increase of approximately 4.5% in the cumulative risk of HRSV infection incidence over a 2-week period. The other meteorological drivers (relative humidity, precipitation, and wind speed) explained approximately 4.8%, 3.7%, and 1.8% of the variation, respectively. Despite these simplified assumptions, our findings objectively reveal that the temporal driving patterns of HRSV infection could be partially explained by these meteorological drivers. In addition to highlighting the relative role played by climate variability, our findings indicate that there may be an additional underlying mechanism involved in shaping the time-dependent transmission heterogenicity of HRSV that remains to be identified. By identifying causal relationships between meteorological drivers and HRSV infection epidemic patterns, we can build predictive models of the epidemic that would help inform guidelines for timing of monoprophylaxis, such as palivizumab, and HRSV vaccination when it becomes available [65,66].

Our results are fairly consistent with the results of previous studies that investigated the broad mechanistic principles underlying the association between meteorological factors and HRSV infection seasonality. For instance, the observation that moderately low ambient temperatures lead to high viral transmission rates has been made in many previous studies. In particular, the negative association between mean ambient temperature and HRSV infection incidence, which was observed in this present study, is biologically valid and consistent with the results of previous studies conducted in mainland China and Brazil [23,67]. Moreover, laboratory studies and animal experiments have described that low ambient temperature plays a potential role in modulating the viability and stability of respiratory viruses by affecting the properties of viral surface proteins and lipid membrane and proportion of droplet nuclei [68,69,70,71]. Lower ambient temperatures may also enhance virus susceptibility by triggering changes in human physiology. In contrast, a linear increase in the cumulative IRR as the mean ambient temperature increased to above approximately 20.0 °C was observed in this study, similar to the form of nonlinear association identified in two studies that were recently conducted in Singapore [50,51]. For instance, HRSV infection epidemics in tropical and subtropical regions (e.g., Hong Kong, Singapore, Malaysia, and Colombia) have been associated with high ambient temperatures [72]. Recently, it has also been observed that the HRSV infection seasonality has changed from autumn–winter (September–February) to summer–autumn (June–November) in Japan after 2017, which is generally a temperate region [6,7,8,9]. A notable study using national Japanese surveillance and meteorological data examined the causes of summer–autumn epidemic conditions and found that the interaction between ambient temperature (≥28.2 °C) and relative humidity (≥79.0%) had a 1.92-fold (95% CI, 1.60–2.23) marginal effect on HRSV infection incidence [9]. Another study by Yusuf et al. that examined HRSV infection incidence throughout a year in nine cities that had different climates and geographical locations suggested that the incidence was related to ambient temperature in a bimodal fashion, with greater disease incidences at ambient temperatures of 24.0–30.0 °C and 2.0–6.0 °C [73]. These accumulated epidemiological findings suggest that HRSVs are partially active at relatively high and low ambient temperatures. Although the underlying cause for this complex form of association is unclear, our study, which captured data over multiple seasons across climate zones in Japan, found that ambient temperature had a U-shaped association with incidence. Our findings provide an important perspective on how the variation in ambient temperature between regions may affect HRSV infection epidemics in Japan.

There is ample evidence of a positive association between humidity (including relative and/or absolute) and HRSV infection epidemics, generally consistent with our results. Studies conducted in several regions, including Mexico, Spain, and Italy, have shown an association between high HRSV infection incidence with high humidity [18,25,73]. Indeed, our results are consistent with the modeling results of a study conducted in Singapore that found an inverse J-shaped association between humidity and HRSV infection epidemics [51]. One potential mechanism by which high humidity may facilitate HRSV transmission is through increased viral survival; as HRSV is transmitted via droplets and respiratory secretions, it tends to be rapidly inactivated in small aerosols at low humidity levels and higher HRSV stability in large-particle aerosols at higher humidity levels [18,51,73,74].

To our knowledge, little is known about the effects of precipitation and wind speed on HRSV dynamics; however, our findings show that high levels of precipitation and low levels of wind speeds are weakly associated with increased HRSV infection risk in Japan. Previously, several studies in Italy, Singapore, Philippine, Thailand, and Brazil have already reported that high precipitation is associated with increased HRSV infection incidence, suggesting that the epidemic seasonality may show a similar pattern to precipitation [51,52,67,75,76]. Indeed, the peak in HRSV infection epidemics coincides with the rainy season in several tropical locations [27,28,29,30,31]. In contrast, limited studies in Singapore and Kenya have found wind speed to be negatively associated with the HRSV infection incidence [32,51]. Generally, some of the previous literature suggests that low wind speeds can prolong the presence of infected aerosol particles in the air, thereby facilitating the transmission of respiratory viruses [77]. Also, higher wind speeds can disperse droplets and decrease the concentration of infected aerosol particles [77,78]. Overall, these epidemiological findings were partially consistent with the trends in our results; however, further studies are needed to assess the relative role of these drivers in the HRSV transmission.

Overall, a possible another interpretation for our observations is that ambient meteorological drivers are associated with the amount of time spent indoors and outdoors, thereby facilitating transmission. Time spent indoors in closed or confined environments could theoretically facilitate the transmission of respiratory viral infections such as HRSV. Indoor congestion has often been hypothesized as a primary contributor to the surge in HRSV infection cases during colder seasons in temperate regions [79,80]. Conversely, in subtropical regions, the prevalence of hot and humid climatic conditions during summer drives individuals to seek shelter in densely populated, air-conditioned settings, thereby creating a favorable environment for viral transmission [81]. Notably, most meteorological conditions vary between indoor and outdoor environments and the form of associations also differs (e.g., influenza and COVID-19) [33,82,83]. For instance, these meteorological drivers may be a proxy for another variable that affects transmission (e.g., human behavior and living environment), or there may be a complex relationship between climate variability and disease transmission (e.g., nonlinear or threshold effects). Indeed, these ambiguities may render complex nonlinear associations more plausible. Despite the regional proximity and similar climates, the epidemiological patterns of HRSV infection in the Western Pacific and South-East Asia regions, including Japan, appear to exhibit subtle differences, suggesting that climatic factors alone do not entirely dictate epidemic seasonality. Indeed, one of our significant concerns is the lack of studies on the association between ambient meteorological drivers, human behavior, and HRSV infection epidemics. Thus, detailed, multicountry, multicity studies with different epidemiological contexts and long-term studies are therefore needed to further disentangle complex interactions and present robust evidence.

There are several caveats when interpreting the results of this study. First, our study was based on the analysis of secondary data and is regarded as a type of ecological study in causal inference, which is prone to unobserved confounding [8,11,53,54]. In particular, we did not have access to data on the different genetic strains of HRSV that emerged over time in the different prefectures; therefore, we did not explicitly model the effects of HRSV subtypes (A and B), which could have helped to explain the differences in the HRSV transmission seasonality in Japan [84]. Furthermore, increased contact structures (i.e., contact patterns and rates) among older students during school terms have been also hypothesized to play an important role in HRSV seasonality [85,86]. These factors influence the accurate modeling of the true extent of disease infections and transmission within populations; however, disease surveillance data allow researchers to determine temporal trends in disease dynamics that can be further modeled and used in public health decision making. In particular, the influence of individual exposure on infection risk and causality at the individual level cannot be inferred over a large geographical area of Japan. Second, our time-series statistical model only accounted for meteorological drivers; other important drivers such as human behavioral changes could also drive viral transmission, although some of these were associated with the selected meteorological drivers (e.g., indoor crowding in hot/cold weather) [24,79,80,81]. Since the beginning of the COVID-19 pandemic, a near cessation of HRSV transmission during periods of movement restrictions and physical distancing has been observed. However, on/off-season HRSV infection epidemics have been reported in multiple countries, which may be attributed to the relaxation of nonpharmaceutical interventions and an increase in population susceptibility [87,88]. As all our data were collected prior to the COVID-19 pandemic, our current modeling framework was not devised to incorporate these factors. Fourth, our study may be limited in its generalizability due to the collection of data from a specific geographical area (47 Japanese prefectures) and a limited period (2014–2019), thus precluding the representation of different epidemic periods across diverse regions. For similar future studies, it may be beneficial to consider increasing the number of geographical regions, expanding the set of indicators, and including additional data, such as individual-level data. Fifth, although we only utilized meteorological drivers such as mean ambient temperature, relative humidity, precipitation, and wind speed as explanatory variables in our time-series statistical models, other environmental confounders such as diurnal temperature ranges, sunshine hours, and ultraviolet levels may also contribute to HRSV transmission [50,51,89,90,91]. In particular, some air pollutants (e.g., particulate matter of aerodynamic diameter of less than 2.5 µm, nitrogen dioxide, sulfur dioxide, and carbon monoxide) have been widely considered in epidemiological studies of HRSV and further modelling studies are needed to quantify the contribution of these potential drivers to the transmission dynamics of HRSV across Japan and to understand the interrelationships [50]. Seventh, because our study focused on pooled associations between meteorological drivers and seasonal variations in incidence, we did not fully characterize the association between the heterogeneous transmission of HRSV in specific Japanese prefectures, which was beyond the scope of this study. For instance, the occurrence in local HRSV infection epidemics in remote prefectures of Japan should be considered, as the epidemiological effect on HRSV transmission may differ among regions [11,46]. To identify the actual causes of the heterogeneity among the prefectures, we need to validate these causes by formulating sophisticated spatiotemporal hierarchical modeling frameworks (e.g., approximate Bayesian inference using integrated nested Laplace approximation) and DLNMs [92] while simultaneously accounting for spatial heterogeneity and autocorrelation or a two-stage time-series design involving a downscaling procedure [37,56]. Moreover, the application of techniques such as analysis of partial differential equation systems and numerical simulation may also allow for more flexible modelling of exposure–lag–response relationships [93,94,95,96]. Indeed, this our method does not provide site-specific information on meteorological relationships with HRSV infection incidence. Therefore, our present results reflect an overall average trend during the study period in Japan that should be interpreted with caution. Finally, although this study successfully described the space-varying, nonlinear, and multidelayed associations between the meteorological drivers and HRSV infection seasonality across Japan, there remain some future studies that could expand our current knowledge. Particularly, the extent to which the future long-term risk of HRSV infection may change under different current and future climate change scenarios has not been predicted and is not sufficiently clear. For instance, harmonized input data and common simulation protocols from the Inter-Sectoral Impact Model Intercomparison Project (ISIMIP) can provide useful insights into the possible consequences of several key climate change options open to us (e.g., dengue, malaria, and vibriosis) [97].

## 5. Conclusions

Disease transmission is a dynamic interplay involving a network of contributing factors. Here, using empirical epidemiological data, we characterized the nonlinear and multidelayed temporal associations between the meteorological drivers and HRSV infection seasonality in the pre-HRSV-vaccination and pre-COVID-19 era across Japan. As a result, our findings indicate that hot and humid weather, high levels of precipitation, and low levels of wind speeds are associated with increased HRSV infection risk in Japan. Note that most HRSV and other seasonal respiratory viruses actually transmit indoors, rather than outdoors; therefore, the relevance of these outdoor climate correlations maybe most useful as a surrogate for indoor environments at the time of such HRSV incidence and transmission events. Our findings align, to an extent, with available mechanistic explanations and reports from the previous literature from other regions of the world. The meteorological drivers investigated in our method may help researchers to partially explain differences in the strength of seasonality across the different regions of Japan, in contrast with previous methods that assume a linear relationship of seasonality with disease risk and changing exposures. More broadly, this study may suggest to policymakers the need for public health strategies and interventions that are flexibly adapted to the climatic conditions in different regions of Japan to mitigate HRSV transmission (e.g., increase the intensity and frequency of interventions in periods of hot and humid weather, high levels of precipitation, and low levels of wind speeds). More importantly, substantively, with several prophylactic vaccine candidates (e.g., prefusion-stabilized F protein vaccine) on the horizon, our present findings on the HRSV infection seasonality in Japan could help to predict the onset of the HRSV season and may inform the timing of immunization as well as seasonal health service planning by pediatric clinical and laboratory diagnostic teams to prevent severe outcomes in high-risk groups [65,66]. Future analyses should consider how the associations addressed in this study can inform the forecasting of the spatiotemporal distribution and seasonality of HRSV under anticipated climate regimes, immunization, and COVID-19-related disruptions.

## Figures and Tables

**Figure 1 viruses-15-01914-f001:**
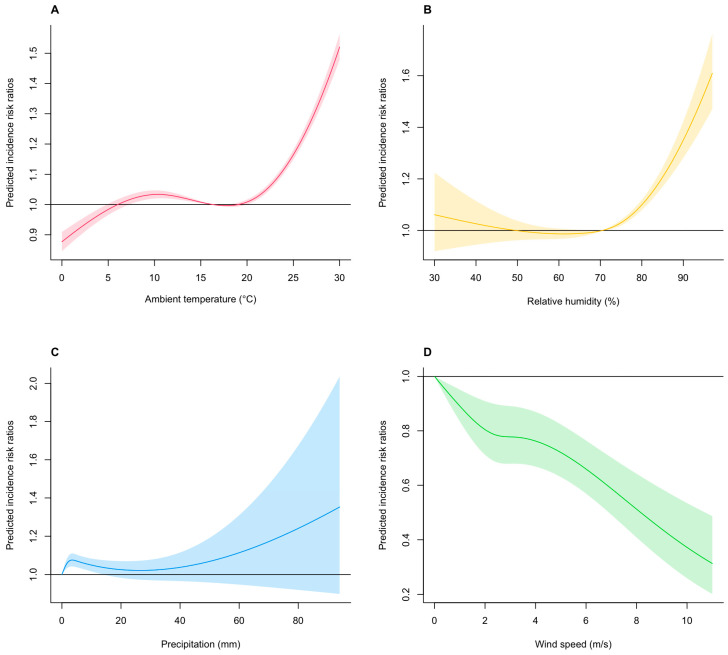
Assessing the pooled nonlinear association of IRRs of HRSV incidence with meteorological variables. (**A**) Overall association of the 2-week cumulative risk of the percent change in the estimated human respiratory syncytial virus (HRSV) infection incidence with weekly mean ambient temperature (unit: °C). (**B**) Overall association of the 2-week cumulative risk of the percent change in the estimated HRSV infection incidence with weekly relative humidity (unit: %). (**C**) Overall association of the 2-week cumulative risk of the percent change in the estimated HRSV infection incidence with weekly precipitation (unit: mm). (**D**) Overall association of the 2-week cumulative risk of the percent change in the estimated HRSV infection incidence with daily weekly wind speed (unit: m/s). The present study covers the period between 1 January 2014 and 29 November 2019 (between the 1st week of 2014 and 52nd week of 2019) across all 47 prefectures in Japan. The red, yellow, blue, and green lines represent the estimated incidence relative risks (IRRs) of HRSV infection, with the shaded bands representing the 95% confidence intervals (CIs). The corresponding reference values are 16.3 °C (**A**), 70.2% (**B**), 0.0 mm (**C**), and 0.0 m/s (**D**). The relevant selected estimates (i.e., IRR and 95% CIs) for this figure are shown in Table 2.

**Table 1 viruses-15-01914-t001:** Descriptive statistics for the number of weekly newly confirmed HRSV cases and meteorological variables.

Potential Drivers	Mean	SD	Min	P_25_	P_50_	P_75_	Max
Weekly newly confirmed cases	49	76	0	7	22	58	892
Mean ambient temperature (°C)	15.7	8.3	–6.7	8.1	16.3	22.7	32.2
Relative humidity (%)	69.5	9.5	30.8	63.1	70.2	76.4	97.2
Precipitation (mm)	4.7	6.4	0.0	0.7	2.9	6.2	94.1
Wind speed (m/s)	2.9	0.9	0.9	2.2	2.8	3.4	11.3

Abbreviations: SD, standard deviation; Min: minimum; P_25_, 25th percentile; P_25_, 50th percentile; P_75_, 75th percentile; Max: maximum. Notes: the present study covers the period between 1 January 2014 and 29 November 2019 (between the 1st week of 2014 and 52nd week of 2019) across all 47 prefectures in Japan.

**Table 2 viruses-15-01914-t002:** Forecasting the specific IRRs of nonlinear associations between the HRSV incidence and meteorological variables.

Potential Drivers	Lag (Weeks)			
	0	1	2	0–2
	IRR (95% CI)	IRR (95% CI)	IRR (95% CI)	IRR (95% CI)
Mean ambient temperature (°C)				
8.1 °C	1.053 (1.015, 1.093)	1.117 (1.070, 1.166)	0.869 (0.837, 0.903)	1.023 (1.008, 1.039)
22.7 °C	0.936 (0.910, 0.964)	1.037 (1.001, 1.073)	1.099 (1.067, 1.131)	1.068 (1.056, 1.079)
Relative humidity (%)				
63.1%	0.998 (0.986, 1.011)	0.979 (0.966, 0.991)	1.009 (0.997, 1.022)	0.997 (0.971, 1.003)
76.4%	1.007 (0.997, 1.017)	1.039 (1.029, 1.050)	0.998 (0.988, 1.008)	1.045 (1.032, 1.059)
Precipitation (mm)				
0.7 mm	1.002 (0.995, 1.008)	1.012 (1.005, 1.019)	1.015 (1.005, 1.019)	1.027 (1.015, 1.038)
6.2 mm	0.983 (0.964, 1.002)	1.033 (1.014, 1.053)	1.047 (1.027, 1.067)	1.064 (1.032, 1.097)
Wind speed (m/s)				
2.2 m/s	0.972 (0.877, 1.078)	0.887 (0.799, 0.985)	0.919 (0.829, 1.019)	0.793 (0.697, 0.903)
3.4 m/s	0.976 (0.875, 1.088)	0.864 (0.774, 0.965)	0.918 (0.824, 1.022)	0.775 (0.678, 0.885)

Abbreviations: IRR, incidence relative risk; CI, confidence interval; HRSV, human respiratory virus. Notes: the present study covers the period between 1 January 2014 and 29 November 2019 (between the 1st week of 2014 and 52nd week of 2019) across all 47 prefectures in Japan. Associations between the HRSV infection incidence and mean ambient temperature (unit: °C), relative humidity (unit: %), precipitation (unit: mm), and wind speed (unit: m/s) are described as IRRs with 95% confidence intervals (CIs) with reference to the IRRs at 16.3 °C, 70.2%, 0.0 mm, and 0.0 m/s, respectively; 8.1 °C and 22.7 °C correspond to the 25th and 75th percentiles of mean ambient temperature, respectively; 63.1% and 76.4% correspond to the 25th and 75th percentiles of relative humidity, respectively; 0.7 mm and 6.2 mm correspond to the 25th and 75th percentiles of precipitation, respectively; 2.2 m/s and 3.4 m/s correspond to the 25th and 75th percentiles of wind speed, respectively.

## Data Availability

An anonymized dataset that enables replication of the analysis is publicly available and is available from the corresponding author upon request.

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
