# Peer review of "Nonlinear and Multidelayed Effects of Meteorological Drivers on Human Respiratory Syncytial Virus Infection in Japan"

_viruses, 2023, doi:10.3390/v15091914_

Round 1

Reviewer 1 Report

This study examined the association between meteorological factors and human respiratory syncytial virus (HRSV) infections in Japan. It is a very interesting study as few Asian studies have looked at the environmental drivers for RSV seasonality. Some comments:

1. Introduction: What is the seasonality of RSV in Japan, compared to other similar settings?

2. Methods: How the maximum number of lags being determined?

3. What is the explanation for the negative association of Wind speed?

4. Any polluants have been accounted in the model?

5. The discussion is too long with too many numbers of irrelevant citations.

Acceptable. Recommend to have a proofread.

Reviewer 2 Report

What was the primary objective of this study, and how did the researchers aim to contribute to the understanding of human respiratory syncytial virus (HRSV) infection epidemics in Japan?

Could you elaborate on the data collection process, including the specific time frame and geographical coverage for the weekly time-series data of newly confirmed HRSV infection cases and meteorological variables from Japanese prefectures?

How did the researchers approach the analysis of the complex relationship between HRSV infection incidence and meteorological drivers? Could you explain the methodology that combined standard time-series generalized linear models with distributed lag non-linear models?

What were the main exposure–lag–response associations observed in relation to the incidence relative risks (IRRs) of HRSV infection and its meteorological drivers? Could you provide insights into the effects of ambient temperature and relative humidity on HRSV infection incidence?

Were there any notable findings regarding the associations between precipitation and wind speed with HRSV infection incidence? How did these meteorological variables contribute to the overall understanding of HRSV transmission?

In the context of the broader temporal landscape, how do these findings contribute to the understanding of the seasonality of HRSV transmission in Japan, particularly in the pre-HRSV-vaccination and pre-coronavirus disease 2019 (COVID-19) era?

What implications might these findings have for public health strategies or interventions related to HRSV infection in Japan? How could the insights gained from this study inform future efforts to mitigate HRSV transmission?

Did the study identify any limitations or areas for further research that could enhance the understanding of the complex interplay between meteorological drivers and HRSV infection epidemics?

How does this study contribute to the broader field of infectious disease epidemiology, especially in terms of its approach to characterizing the non-linear and multi-delayed effects of meteorological factors on disease dynamics?

Considering the significance of this research, what are the potential avenues for future studies or investigations that could build upon these findings and expand our knowledge of HRSV infection dynamics in response to meteorological influences?

Add in bibliography

Analysis and numerical simulation of system of fractional partial differential equations with non-singular kernel operators

Fractional View Analysis of Emden-Fowler Equations with the Help of Analytical Method

Investigating Families of Soliton Solutions for the Complex Structured Coupled Fractional Biswas–Arshed Model in Birefringent Fibers Using a Novel Analytical Technique

Probing Families of Optical Soliton Solutions in Fractional Perturbed Radhakrishnan–Kundu–Lakshmanan Model with Improved Versions of Extended Direct Algebraic Method

please check grammatical mistakes by grammeral software 

Round 2

Reviewer 2 Report

Accept